# Dental Pulp Stem Cell-Derived Factors Alleviate Subarachnoid Hemorrhage-Induced Neuroinflammation and Ischemic Neurological Deficits

**DOI:** 10.3390/ijms20153747

**Published:** 2019-07-31

**Authors:** Te-Fu Chen, Kuo-We Chen, Yueh Chien, Ying-Hsiu Lai, Sung-Tsang Hsieh, Hsin-Yi Ma, Kou-Chung Wang, Chia-Yang Shiau

**Affiliations:** 1Department of surgery, Division of Neurosurgery, National Taiwan University Hospital, Taipei 100, Taiwan; 2Graduate Institute of Medical Sciences, National Defense Medical Center, Taipei 114, Taiwan; 3Department of Neurosurgery, Tri-Service General Hospital, Taipei 115, Taiwan; 4Non-invasive Cancer Therapy Research Institute - Taiwan, Taipei 104, Taiwan; 5Institute of Pharmacology, School of Medicine, National Yang-Ming University, Taipei 112, Taiwan; 6Department of Medical Research, Taipei Veterans General Hospital, Taipei 112, Taiwan; 7Department of Anatomy and Cell Biology, College of Medicine, National Taiwan University, Taipei 102, Taiwan

**Keywords:** aneurysmal subarachnoid hemorrhage, stem cell, neuroinflammation, conditioned medium

## Abstract

Aneurysmal subarachnoid hemorrhage (aSAH), characterized by the extravasation of blood into the subarachnoid space caused by an intracranial aneurysm rupture, may lead to neurocognitive impairments and permanent disability and usually carries poor outcome. Dental or gingiva-derived stem cells have been shown to contribute to immune modulation and neuroregeneration, but the underlying mechanisms are unclear. In the present study, we sought to investigate whether dental pulp stem cells (DPSCs) secrete certain factor(s) that can ameliorate the neural damage and other manifestations in a rat aSAH model. Twenty-four hours after the induction of aSAH, microthrombosis, cortical vasoconstriction, and the decrease in microcirculation and tissue oxygen pressure were detected. Intrathecal administration of DPSC-derived conditioned media (DPSC-CM) ameliorated aSAH-induced vasoconstriction, neuroinflammation, and improved the oxygenation in the injured brain. Rotarod test revealed that the aSAH-induced cognitive and motor impairments were significantly improved by this DPSC-CM administration. Cytokine array indicated the major constituent of DPSC-CM was predominantly insulin growth factor-1 (IGF-1). Immunohistochemistry staining of injured brain tissue revealed the robust increase in Iba1-positive cells that were also ameliorated by DPSC-CM administration. Antibody-mediated neutralization of IGF-1 moderately deteriorated the rescuing effect of DPSC-CM on microcirculation, Iba1-positive cells in the injured brain area, and the cognitive/motor impairments. Taken together, the DPSC-derived secretory factors showed prominent therapeutic potential for aSAH. This therapeutic efficacy may include improvement of microcirculation, alleviation of neuroinflammation, and microglial activation; partially through IGF-1-dependent mechanisms.

## 1. Introduction

Aneurysmal subarachnoid hemorrhage (aSAH), characterized by the extravasation of blood into the subarachnoid space caused by an intracranial aneurysm rupture, is an emergency medical condition throughout the world. aSAH commonly presents as a severe headache that is often associated with nausea, vomiting, neck rigidity, and photophobia. In addition, depending on its severity, aSAH may also exhibit symptoms including drowsiness, confusion, focal neurological deficits, hemiparesis, and coma [1]. Current treatments for aSAH-related vasoconstriction include Triple-H therapy (Hemodilution, Hypertension, and Hypervolemia), in order to keep adequate cerebral perfusion, and initial surgical approaches such as craniotomy aneurysmal clipping or endovascular repairs aim to prevent rebleeding. Unfortunately, many survivors still suffer from long-term physical, neurocognitive, psychiatric, and/or psychological impairments [1,2]. Factors that contribute to the poor outcome in aSAH patients include the initial volume of hemorrhage, re-bleeding, and the degree of cerebral ischemia [3,4]. Recent data suggested that neuroinflammation may play a crucial role in the process of brain injury expansion and neurodeficits following aSAH [5]. Therefore, it is urgently needed to develop a therapeutic approach that can target neuroinflammation and effectively rescue patients from aSAH-related abnormalities, including diffuse vasospasm and delayed neurological deficits.

Stem cell-based therapy that holds promising potential in regeneration medicine has been considered as an ideal approach to repair tissue damages. In animal experiments, a variety of stem cells, including embryonic stem cells (ESCs), neural stem cells (NSCs), induced pluripotent stem cells (iPSCs), and mesenchymal stem cells (MSCs) have been employed for treating neurological deficits. For example, mouse ESCs have been shown to ameliorate ischemic stroke induced by middle cerebral artery occlusion [6,7]. Another study showed that both adult and ESC-derived neural precursor cells improved the behavioral dysfunction in rats with ischemic stroke [8]. It has also been shown that transplantation of iPSCs also reduced infarct size and neuronal damage after ischemic stroke [9]. However, cell therapy using ESCs or iPSCs usually carries several problems, including ethical consideration and the risk of tumor formation. Mesenchymal stem cells (MSCs) have been found within various tissues, including the adipose tissue, bone marrow, hair follicles, the umbilical cord, and dental pulp. Among all of these MSCs, dental pulp has been shown to contain multipotent progenitors and pluripotent stem cells and can be further differentiated into neural-like cells that may be beneficial to neural repair [10]. Nevertheless, whether MSCs also possess therapeutic efficacy on the neuroinflammation and brain damage of aSAH remained uncertain.

MSCs have been shown to carry promising potential in several clinical diseases [11,12]. However, since implanted MSCs do not survive for long, it is widely believed that the bioactive factors produced by MSCs exert beneficial therapeutic effect and contribute to the MSC-mediated beneficial effects in synergism [13,14]. Actually, MSC-conditioned medium has been shown to provide effective cell-free therapy against inflammatory arthritis [15] and focal cerebral ischemia reperfusion injury [16], as well as neural trauma and stroke [13]. In brain injury, the bioactive substances secreted by MSCs have been reported to promote neurogenesis, angiogenesis, and an anti-inflammatory effect [17,18,19], and may eventually lead to the stabilization of microenvironment/metabolic profiles in the injured brain. Recently, we established a rat experimental aSAH model that recapitulates neural damage and impaired microcirculation of aSAH [20]. In the present study, we want to isolate rat dental pulp-derived stem cells (DPSCs) and examine whether such DPSCs can secrete some bioactive substances that can ameliorate neuroinflammation and restore the dys-regulated cerebral microcirculation and cognitive/behavior impairments in rats with aSAH.

## 2. Results

### 2.1. Preparation of Dental Pulp-Derived Mesenchymal Stem Cells

Dental pulp, the soft central part of a tooth that can be easily collected from extracted teeth, has been shown to serve as a reservoir of dental pulp stem cells (DPSCs). Prior to examining the therapeutic potential of DPSC-derived conditioned media (DPSCs-CM) on aSAH, we extracted incisor teeth and isolated DPSCs from the pulp tissue within the teeth of Wistar rats (Figure 1A, top. Microscopic examination revealed that DPSCs exhibited fibroblast-like morphology (Figure 1A, bottom). As analyzed by flow cytometry, DPSCs were highly enriched for CD44, CD90, CD105, and CD73 (Figure 1B). These mesenchymal features and morphology illustrated the characteristics of DPSCs.

### 2.2. The Conditioned Medium of Cultivated DPSCs Ameliorated aSAH in Rats

Recently, we established an experimental aSAH model that recapitulates neural damage and impaired microcirculation in rats [20]. After the craniotomy and dura opening, we used a dissection microscope equipped with a capillary videoscope to monitor the cortical microcirculations. With this setup, the cortical microcirculations including the main arterioles and venules could be clearly observed. Around one to two main arterioles could be found in the craniotomy site, and each arteriole was branched into primary arterioles (pa), secondary arterioles (sa), and terminal arterioles (ta) (Figure 2). After Twenty-four hours of the aSAH induction, significant alterations in microcirculation over the brain surface were observed in the experimental animals with craniotomy (Figure 2, middle panel). Compared with the arterioles in the sham group, diffuse vasoconstriction was observed in the secondary and terminal arterioles in the aSAH group (Figure 2, middle panel). To examine whether DPSC-CM could rescue experimental aSAH, we collected the conditioned medium from the culture of DPSCs and applied the injection of DPSC-CM via the foramen magnum of rats at the onset of aSAH induction. Microcirculation imaging in the aSAH group with DPSC-CM administration revealed a significant amelioration of diffuse vasoconstriction (Figure 2, right panel). To validate the therapeutic potential of DPSC-CM, the artery blood pressure (mean arterial pressure, MAP), microcirculation, cortical blood flow, and brain oxygen pressure in rats receiving the indicated treatment were measured. Neither induction of aSAH nor addition of DPSC-CM affected the artery blood pressure (MAP) (Figure 3A). Administration of DPSC-CM per se did not modify the diameter of arterioles of any size (primary, secondary, and terminal arterioles; Figure 3B). Induction of aSAH did not show a detectable effect on primary arterioles, but significantly reduced the diameter of secondary and terminal arterioles (Figure 3B). Remarkably, the reduction in the diameter of secondary and terminal arterioles was rescued by the 24 h exposure to DPSC-CM (Figure 3B). Induction of aSAH largely reduced cortical blood flow and brain oxygen pressure. This reduction in blood flow was substantially rescued by the co-administration of DPSC-CM, while the oxygen pressure was moderately improved by the same treatment (Figure 3C,D). Nevertheless, administration of DPSC-CM alone showed no effect on cortical blood flow and brain oxygen pressure (Figure 3C,D).

### 2.3. IGF-1 as the Major Active Constituent in the DPSC-CM

To elucidate the active constituents in the DPSC-CM, we subjected the DPSC-CM to Quantibody^®^ Cytokine Arrays, the multiplex ELISA antibody arrays, for screening and comparing the expression levels of various cytokines. The total protein concentration of DPSC-CM was 27 (±2.1) μg/mL. The active constituents that were found in the DPSC-CM included insulin-like growth factor 1 (IGF-1), tissue inhibitor of metalloproteinases 2 (TIMP2), tissue inhibitor of metalloproteinases 1 (TIMP1), transforming growth factor β (TGF-β), and others. Among these active constituents, the level of IGF-1 was the highest and accounted for 42.64% of total protein in the DPSC-CM (Figure 4).

### 2.4. DPSC-CM Alleviates Neuroinflammation in aSAH Partially via IGF-1

In this study, we therefore examined whether neuroinflammation was involved in this experimental aSAH model, and whether DPSC-CM-mediated rescuing effects were associated with alleviating neuroinflammation. Twenty-four hours after aSAH induction and the administration of DPSC-CM via foramen magnum, the injured brain tissue was subjected to immunohistochemistry analysis. The marker (Iba-1) for activated microglia was used for examining the involvement of neuroinflammation in the injured brain. Immunohistochemistry with Iba-1 revealed that Iba-1 signal was rarely detected in control sham rats (Figure 5A,E), whereas Iba-1 signal was substantially increased over the injured brain (Figure 5B,F), indicating diffuse microglial activation and neuroinflammation in aSAH rats. Importantly, treatment of DPSC-CM for 24 h also exhibited therapeutic potential and effectively reduced the amount of Iba-1-positive cells in aSAH rats (Figure 5C,G).

Considering the substantial amount of IGF-1 in DPSC-CM, we next thought to determine the involvement of IGF-1 in the DPSC-CM efficacy that ameliorates microglial activation and neuroinflammation in aSAH. aSAH-injured rats were treated with neutralizing antibodies against IGF-1 and DPSC-CM for 24 h. Immunohistochemistry of Iba-1-positive cells revealed that IGF-1 neutralization partially blunted the efficacy of DPSC-CM (Figure 5D,H). Collectively, these data indicated that in the established aSAH model, the pathology of aSAH involved neuroinflammation that can be alleviated by DPSC-CM. Remarkably, IGF-1, the major constituent of DPSC-CM, appeared to partially contribute to this DPSC-CM-mediated neuroprotective effect.

### 2.5. DPSC-CM Improves Microcirculation in aSAH Partially via IGF-1

Our data revealed that IGF-1 is the most abundant bioactive constituent in DPSC-CM, and antibody-mediated neutralization of IGF-1 moderately abrogated the DPSC-CM-mediated effect on microglial activation/neuroinflammation (Figure 5I). Microcirculation imaging revealed that aSAH led to severe and diffuse vasoconstriction, and DPSC-CM consistently ameliorated such impaired microcirculation (Figure 6A–C). Remarkably, upon the administration of IGF-1-neutralizing antibody, the vasoconstriction was partially retained in the secondary and terminal arterioles despite the presence of DPSC-CM (Figure 6D). The artery blood pressure (MAP), arteriole diameter, cortical blood flow, and brain oxygen pressure in all groups were subsequently measured. No obvious effect in artery pressure was observed among all Sham-operated or aSAH-injured rats, with or without all given treatment (Figure 6E). Since both aSAH and DPSC-CM did not induce significant change in primary arteriole diameter (Figure 3B), we focused on the diameter change of the secondary and terminal arterioles (Figure 6F). aSAH led to an expected decrease in the diameters of the secondary and terminal arterioles that were restored by DPSC-CM (Figure 6F), and administration of IGF-1 antibody moderately neutralized this DPSC-CM effect in the secondary and terminal arterioles (Figure 6F). Meanwhile, administration of IGF-1-neutralizing antibody also moderately blunted the cortical blood flow and brain oxygen pressure in the presence of DPSC-CM in aSAH-injured rats (Figure 6G,H). Collectively, these data indicated that IGF-1 is partially responsible for the beneficial effect of DPSC-CM that rescues impaired microcirculation in aSAH-injured rats.

### 2.6. Therapeutic Efficacy of DPSC-CM on Cognitive and Motor Impairments in aSAH-Injured Rats

aSAH has been characterized by the hemorrhage into the intracranial aneurysm that may lead to several neurological deficits, including neurocognitive impairments and permanent disability. Therefore, it is a critical issue if administration of DPSC-CM also improved the neurocognitive and motor impairments in our established aSAH model. The Rotarod test was used to evaluate the motor coordination and neurocognition of sham-operated or aSAH-injured rats with all indicated treatment (Figure 7A). aSAH-injured rats exhibited cognitive impairment and motor dysfunctions, as compared with Sham-operated rats. Importantly, DPSC-CM treatment almost fully restored the cognitive and motor functions in aSAH-injured rats (Figure 7B). Antibody-mediated neutralization of IGF-1 moderately abrogated the DCSC-CM effect observed in the Rotarod test (Figure 7B), indicating that IGF-1 also served a role in the DPSC-CM-induced restorative effect on neurocognitive and motor functions in aSAH rats.

## 3. Discussion

According to our previous study, the aSAH model demonstrated that the pathology of impaired microcirculation includes vasoconstriction of cortical arterioles and diffuse thrombosis of capillaries [20]. Superfusion of cerebrospinal fluid (CSF) from both aSAH rats and aSAH patients exhibited vasoconstrictive effects on the brain surface of control rats, indicating that noxious substances in the CSF contributed to the microcirculatory changes and the subsequent progressive ischemic damage [20]. Unexpectedly, using vasodilators that can improve vascular constriction failed to improve outcomes of aSAH patients, suggesting that vasospasm might not account for all the comorbidities of delayed deterioration associated with vasospasm (DDAV) in aSAH. Importantly, it had been proposed that neuroinflammation may serve as the driving force behind the pathology of aSAH that leads to brain injury, vasospasm, and subsequent ischemic brain injury [21,22]. Microglia as the major immune defense scavenging cells are responsible for the clearance of damaged neurons, debris or pathogens in the central nervous system. Such cells have also been known to express specific marker Iba1 (ionized calcium-binding adapter molecule 1) that can be further upregulated by the activation of microglia [23]. Activated microglia can further produce nitric oxide, proteases, hydrogen peroxide and cytokines [24], and usually participate in various noxious events including neuroinflammation, cerebral ischemia, and traumatic brain injury [25]. In the present study, we used Iba1 as the biomarker for microglial activation and neuroinflammation as previously described [25], and delineated that the DPSC-CM-mediated efficacy was associated with amelioration of microglial activation and neuroinflammation.

Stem cell-based therapy had been advocated for the treatment of clinical diseases for years. However, some disadvantages hindered the bioavailability and applications of stem cell-based therapy [26]. Embryonic stem cells (ESCs) are established from cells collected from the inner mass of human embryos at early stages, so that the ethical issues in stem cell-based therapy using ESCs has been argued for a long time [27]. Induced pluripotent stem cells (iPSCs) can be generated from somatic cells via transducing reprogramming factors (e.g., Oct3/4, Sox2, Klf4, and c-Myc). Similar to ESCs, iPSCs also exhibit high pluripotency and can differentiate into various functional types of cells. However, the risk of teratoma formation after transplantation limits the clinical usefulness of both ESCs and iPSCs [27,28]. In addition, it has also been reported that genomic instabilities and epigenetic variations of iPSCs may substantially affect their properties and utilities [29]. Comparing with ESCs and iPSCs, mesenchymal stem cells (MSCs) have several advantages, including high bioavailability and ease of harvesting, abilities for multilineal differentiation, immunosuppressive effects, and the lack of ethical issues. Several advantageous effects and benefits of MSCs have been reported, raising the possibility that MSCs may show promising potential in clinical diseases.

Transplantation of MSCs for cell therapy has shown significant efficacy in neurodegenerative diseases including Parkinsonism and dementia [11,12]. But the therapeutic potential of MSCs on aSAH is still unknown. In clinical manifestations of aSAH, the initial hemorrhage is believed to elicit a series of pathological events including vasospasm of cerebral vessels and detrimental neurological outcomes. It will be urgently critical to elucidate whether early intervention in the aSAH pathogenesis may effectively ameliorate the impaired cortical circulation/neurological deficits and lead to better treatment outcome. Short replicative lifespan and limited proliferation ability in culture may hinder the bioavailability of MSCs at the acute phase of aSAH. We therefore attempted to evaluate the bioavailability of MSC-derived conditioned media and examine whether administration of such media at the acute phase of aSAH can alleviate the severity and progression of aSAH-associated comorbidities. In this study, using our established experimental model that recapitulates clinically relevant features of aSAH (i.e., impaired microcirculation vasculature, alterations of cerebral blood flow, and brain tissue oxygenation), our data showed that the conditioned media from dental pulp-derived MSC culture can effectively restore the impaired microcirculation/blood perfusion, tissue oxygenation, and ameliorate neuroinflammation within 24 h after the aSAH induction (Figure 8). These data provided evidence showing that early administration with MSC-derived bioactive substances at the onset of aSAH may potentially prevent the deterioration of impaired microcirculation and neuroinflammation and the loss of regular cognitive/motor functions. 

The behaviors of MSCs are various, such as self-renew, differentiation, migration, even in aging, and are accompanied with different compositions of paracrine for different purposes. The protocol of MSC-CM generation must be maintained in the same stage, with similar potential as one of the quality controls [14]. In this study, we made a purification of the constitutes in DPSC-CM with staged Tangential Flow Filtration in order to narrow down the spectrum of proteins in the DPSC-CM. Instead of using all the elements inside the DPSC-CM, we treated the aSAH rats with a relatively purified and quantified protein in order to delineate the anti-inflammatory effects.

Previous study delineated how IGF-1 acts on vessels in injured neointima and tunica media in carotid stenosis. IGF-1 not only played key role of proliferative or anti-apoptotic benefits, but also stabilized the endothelial progenitor cells in the carotid stenosis model [30]. In our study, antibody-mediated IGF-1 neutralization partially reversed the beneficial effect of MSC-derived conditioned media on microcirculation and tissue oxygenation, neuro-inflammation, and cognitive/motor functions, indicating that IGF-1 substantially contributed to the treatment effect induced by the conditioned media in aSAH-injured brain. Consistent with our observations of IGF-1 in the injured brain, Tang X et al. also reported that MSCs promote the recovery of brain function through the release of IGF-1 [31]. IGF-1 has been known to have diverse functions in the central nervous system, including the regulation of early brain development, myelination, the formation of synapses, neurogenesis, and cognition [32,33]. IGF-1 has been extensively accepted to exert wide-spectrum neuroprotection against neuroinflammation and oxidative stress in the brain [34]. Although some controversial data of IGF-1 have been reported in Alzheimer disease and age-related diseases [35,36], the administration of IGF-1 or IGF-1 mimetics still appears to be beneficial when IGF-1 resistance occurs in these diseases [34]. Our immunohistochemistry data (Figure 5) supported that IGF-1 exhibits neuroprotective effects and alleviates neuroinflammation in aSAH. Although IGF-1 has been postulated to induce peripheral arterial vasodilation, the influence of IGF-1 on brain microcirculation vasculature was unclear. Neuroinflammation in aSAH has been proposed to elicit brain injury, vasospasm, and subsequent brain ischemia and injury [21,22], it is plausible that improvements of cortical microcirculation, tissue oxygenation, as well as cognitive/motor functions are secondary to the global alleviation of neuroinflammation by IGF-1 in the aSAH-injured brain. Given that multiple and disseminated cerebral penumbra or ischemia may lead to the clinical neurological deficits, the restoration of cortical microcirculation and tissue oxygenation appear to be the key determinant that contributed to the improvement of motor functions after the DPSC-CM treatment [37]. In this study, the improvement in brain tissue oxygenation and blood flow had correlation to the motor improvement after DPSC-CM/IGF-1 treatment. Taken together, our data suggested that early intervention with DPSC-CM might give better clinical outcome due to the improvement of cerebral ischemia and amelioration of neuroinflammation in the aSAH-injured brain.

In our data of multiplex ELISA antibody arrays, TIMP2, TIMP1, and TGF-β, etc., were also identified in the conditioned media of MSCs. A synergistic effect contributed to by all the cytokines from DPSC found in our data is possible; in particular, TIMP2 exists at a high level in DPSC-CM and is critical to the maintenance of tissue homeostasis. TIMP2 is an inhibitor for matrix metalloproteinase (MMP)-3 and 9, which have been shown to be highly related to a higher risk of soft tissue trauma, such as Achilles tendinopathy [38]. As for functional study of TIMP2 in the neural system, it was remarkably addressed to be beneficial for noise-related cochlear injury [39]. Further studies will be required to elucidate the contribution of these bioactive constituents to the conditioned media-associated beneficial effect in the established aSAH model.

## 4. Materials and Methods

### 4.1. Experimental Model of aSAH

This study was approved by the Animal Committee of National Taiwan University College of Medicine (IACUC approval No:20170435, 3 January 2018). All experiments and protocols were conducted in compliance with human and animal ethical regulations.

The induction of an experimental model for SAH was conducted as previously described [20]. Eight weeks old male Wistar rats (250 to 300 g) were used. All surgical procedures were performed according to the National Institutes of Health Guide for the Care and Use of Laboratory Animals, and were approved by the Committee. The anesthetic was 2.5% isoflurane with 70% nitrous oxide and 27.5% oxygen. It was administered via a tracheostomy tube to ensure deep sedation, as verified by an absence of hind limb and forelimb pain reflexes as well as the absence of corneal reflexes. Normal, nonlabored breathing was maintained throughout the surgery. Temperature was monitored with a rectal probe, and body temperature was maintained at 37 °C with the animal on a thermal blanket. Blood pressure was monitored and maintained at 100–120 mmHg.

The rat aSAH model was performed by following a standard procedure, as described previously [20]. aSAH was induced by injecting 0.3 mL of fresh autologous blood into the cisterna magna over a 5 min period, with the animal in a 20 degree sign head-down position.

### 4.2. Cranial Window for Observation of Microcirculation

Anesthetized rats were placed in a stereotaxic frame (Kopf Instruments Tujunga, Tujunga, CA, USA). After a dorsal scalp incision and blunt dissection to remove loose connective tissue, a cranial window 5 × 5 mm^2^ was made using a saline-cooled drill at the left frontal suture and 1 mm lateral to bregma. The dural matter was opened carefully, and hemostasis was achieved by packing with gauze under a microscope. The diameter of cortical arterioles was quantified by using a CAM1 capillary Anemometer (KK Technology, UK), which has a high resolution (752 × 582 pixels). Monochrome charge-coupled device (CCD) video camera was used for quantification of brain surface microcirculation [20]. All the vessels in the field of craniotomy were taken into account. We firstly identified the main arterioles, and then took photographs along the arterioles and all the branches until communication with the territory of another main arteriole. Then, we sampled all vessels in that territory and identified them as main arterioles, primary artery (pa), secondary artery (sa), and terminal artery (ta), and then measured the diameters of the sampled arterioles individually. The definition and detailed method were written in a previous report [20].

The extent of tissue perfusion and PbtO_2_ in the brain cortex were measured using an OxyLite 2000E detector and an OxyFLO 2000E detector (Oxford Optronic Ltd., Abingdon, England) [40]. The probe was fixed on a stereotactic frame (Stoelting, Wood Dale, IL, USA), and the tip was inserted to a depth of 2 mm underneath the brain surface to start recording.

### 4.3. DPSC Isolation and Culture

Dental pulps were extracted from male Wistar rats using a syringe needle and were transferred into a 25 cm^2^ flask (Corning). Then, dental pulps were cultured in alpha-MEM with 10% FBS (Gibco). Isolation of DPSCs were not subjected to any type of depletion techniques, and when they reached confluence, cells were detached by trypsin (Sigma) and sub-cultured in a 75 cm^2^ flask (Corning) at the density of 2 × 10^3^ cells/cm^2^, and then sub-cultured in a 150 cm^2^ flask (Corning) at a density of 5 × 10^4^ cells/cm^2^ DPSCs at passage 3. The cells were incubated at 37 °C in an atmosphere containing 5% CO_2_ at 100% humidity. The DPSCs used in this study exhibited a fibroblastic morphology, with a bipolar spindle shape, expressed MSC markers (CD44, CD90, CD73, and CD105), but not endothelial/hematopoietic markers such as CD34 and CD45.

### 4.4. Preparation of Conditioned Medium

Conditioned medium (CM) was generated as follows: 80% confluent passage 3–5 DPSCs in a 150 cm^2^ cell culture flask (Corning) were fed with 10 mL DMEM/F12 and 10 mL PBS in each flask, then incubated for 48 h. The medium was collected and centrifuged for 3 min at 2500 rpm. In order to collect proteins with molecular weights between 5 and 30 kDa, CM was collected in a sterilized beaker and further concentrated by using a TFF membrane filter system (Millipore, Burlington, MA, USA) with a 5 kDa and 30 kDa cut-off unit (Millipore) alternatively for 3 h, following the manufacturer’s instructions. The protein concentration in DPSC-CM was measured using a BCA Protein Assay Kit (Pierce, Rock-ford, IL, USA) and adjusted to 27 ± 2.1 μg/mL with Normal saline. Quantitation of each constituent was done after TFF membrane filtration by using Qubit Fluorometric Quantification and Raybiotech ELISA Array.

### 4.5. Treatment Algorithm

Animals were divided into four groups: A: sham operation, B: aSAH with intrathecal injection of normal saline, C: aSAH with intrathecal injection of DPSC-CM, D: aSAH with intrathecal injection of DPSC-CM and anti-IGF-1 protein PEPROTECH (500-P11). There were six rats in each study group. DPSC-CM 20 μL was injected into cisterna magnum 10 min before aSAH, while normal saline was injected in sham treatment groups. Anti-IGF-1 protein 100 μg was mixed with 3 mL CM for each experiment.

### 4.6. Brain Preparation for Immunohistochemistry Analysis

Rats were sacrificed after 24 h of aSAH induction by injection of pentobarbital (200 mg/Kg, i.p.) and perfused trans-cardially with 50 mL of saline followed by 500 mL of a fixative containing 4% paraformaldehyde in 0.1 M phosphate-buffered saline (PBS), pH 7.3 for 30 min. Serial coronal brain sections of brain cortex, 6 mm thick, were cut on a cryostat, thaw-mounted onto gelatin-coated slides, and were used for immunohistochemical staining.

The samples were de-paraffinized by heating at 60 °C for 30 min and xylene. They were then rehydrated by passing through a series of decreasing concentrations of ethanol (100%, 90%, 70%, and 50%) for 5 min each step, and then washed with 0.1 M phosphate-buffered saline. Endogenous peroxidase was quenched with 3% hydrogen peroxidase for 10 min. The sections were incubated with 5% Bovine serum albumin for 1 h to block nonspecific background staining. Subsequently, they were incubated with primary antibody overnight at 4 °C and visualized using the Novolink Polymer Detection System. Iba-1-positive cells were used as a direct indicator of neuroinflammation. The following immunoreagents were used in this study: Anti-Iba-1 Polyclonal Antibody (019–19741; Wako), Novolink Polymer Detection System (RE7140-K; Novocastra, Newcastle upon Tyne, UK).

### 4.7. Rotarod Test

The rotarod test was performed as described by Vogel et al. with small modifications [41]. The animals were trained for three consecutive days at the speed of 4 rpm, three sessions per day for 5 min. If a mouse fell during the habituation period, it was placed back on the instrument. One week after the induction of aSAH, all experimental rats with indicated treatment were assigned for the rotarod test. After the rats were placed on the instrument (Panlab Rota Rod, Havard Apparatus) moving at the speed of 4–40 rpm, in 600 s, the accelerating mode was started (maximum speed 40 rpm). The latency to fall was measured during the 5 min test session.

### 4.8. Statistical Analysis

The results are expressed as mean ± SD. Statistical analysis was performed by two-way ANOVA. Statistical significance is defined as *p* < 0.05.

## 5. Conclusions

In conclusion, we found that dental pulp-derived MSCs released various bioactive constituents into the conditioned media in the culture, and early administration of such media at the onset of aSAH induction effectively alleviated the deterioration of microcirculation, tissue oxygenation, and neuroinflammation in the aSAH-injured brain. Among all identified bioactive constituents in the conditioned media, IGF-1 was the most abundant and significantly contributed to the beneficial effects induced by the MSC-derived conditioned media. Given that IGF-1 has been extensively delineated as a neuroprotective/anti-inflammatory factor, despite some controversial data being reported, alleviation of neuroinflammation appears to be the crucial step for the effective treatment of aSAH. With our established experimental model with clinically relevant characteristics of aSAH [20], our data has implied that early intervention is critical for the therapeutic outcome for aSAH, and therapeutics using MSC-derived bioactive constituents may be a novel and alternative option.

## Figures and Tables

**Figure 1 ijms-20-03747-f001:**
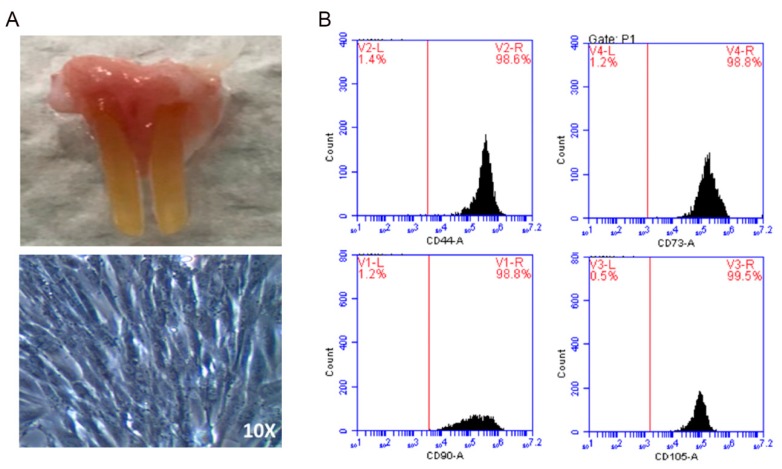
Characterization of rat-derived DPSCs. (**A**) Incisor teeth were extracted from Wistar rats for the isolation of DPSCs (upper). Microscopic examination shows the fibroblast-like morphology of DPSCs after the cultivation onto culture plates (lower). (**B**) Flow cytometry analysis showed the enrichment of CD44, CD90, CD105, and CD73 in DPSCs. DPSCs: dental pulp-derived stem cells.

**Figure 2 ijms-20-03747-f002:**
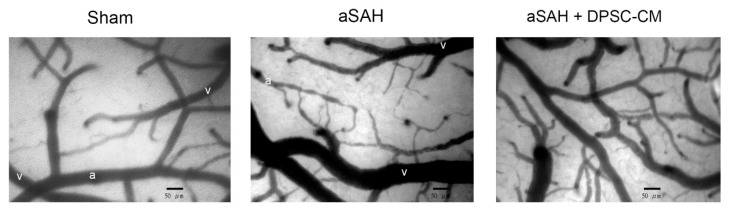
Amelioration of experimental aSAH by foramen magnum injection of DPSC-CM. Rat aSAH model was induced by the injection of fresh autologous blood into the cisterna magna. Induction of aSAH led to a diffuse vasoconstriction, predominantly in secondary and terminal cortical arterioles (middle). Administration of DPSC-CM via foramen magnum at the onset of aSAH induction effectively ameliorated the diffuse vasoconstriction (right). aSAH: aneurysmal subarachnoid hemorrhage. DPSC-CM: DPSC-derived conditioned media. “a”: artery. “v”: vein.

**Figure 3 ijms-20-03747-f003:**
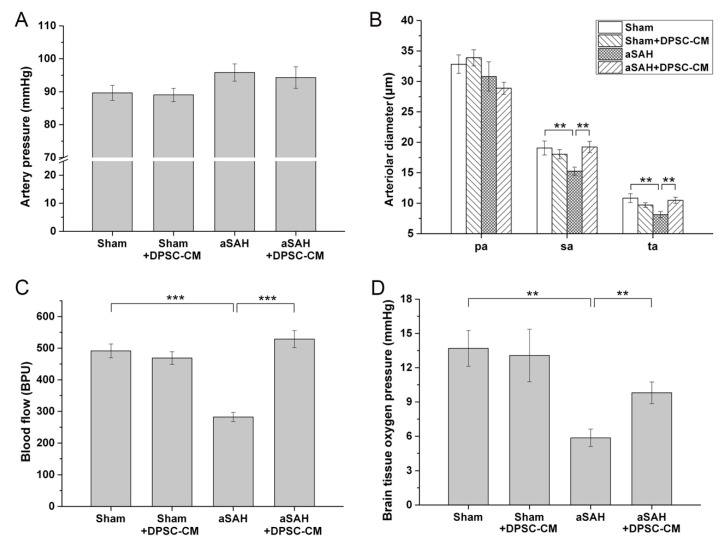
Improvement of parameters of aSAH rat under DPSC-CM treatment. The aSAH model was induced, and DPSC-CM was injected via the foramen magnum of rats at the onset of aSAH induction. After 24 h, the image of microcirculation was captured, and the arteriole diameters of indicated arteriole were quantified. (**A**) Variation of blood pressure (mean arterial pressure, MAP), (**B**) Alterations in arteriole diameter of primary (pa), secondary (sa), and terminal (ta) arterioles in the brain microcirculation, as measured by using CAM1 capillary Anemometer, (**C**) Cortical blood flow, (**D**) Brain oxygen pressure of rats with aSAH induction with or without treatment with DPSC-CM. ** *p* < 0.01, *** *p* < 0.001.

**Figure 4 ijms-20-03747-f004:**
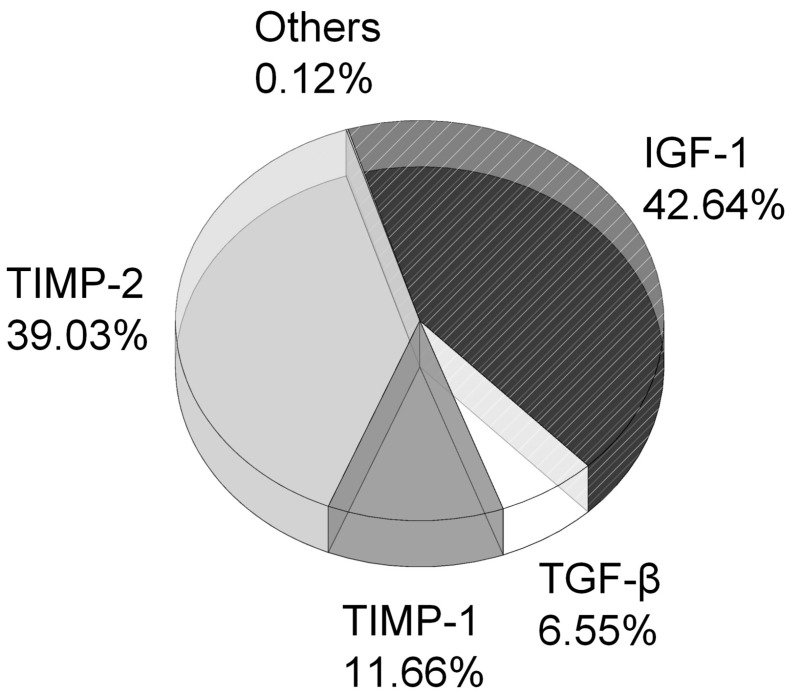
Major active constituents detected in the DPSC-CM. DPSC-CM was subjected to the multiplex ELISA antibody arrays for screening and comparing the expression levels of various cytokines. The percentage of each active constituent in DPSC-CM was calculated and shown. IGF-1: insulin-like growth factor 1. TIMP-2: tissue inhibitor of metalloproteinases 2. TIMP-1: tissue inhibitor of metalloproteinases 1. TGF-β: transforming growth factor β.

**Figure 5 ijms-20-03747-f005:**
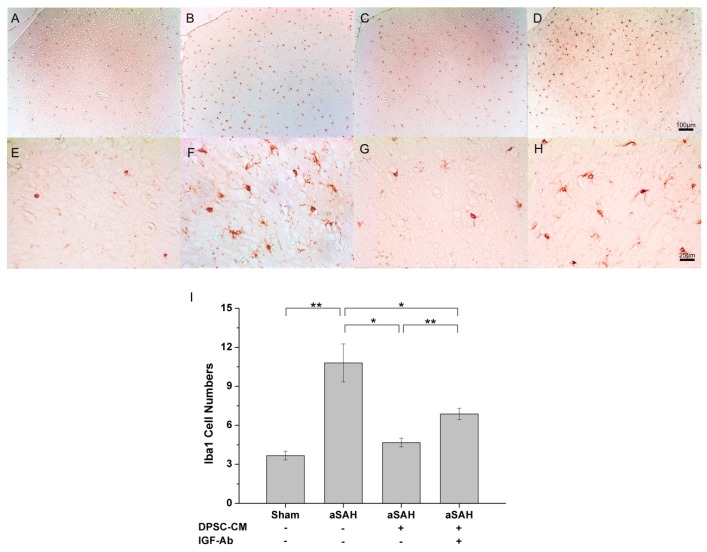
Alleviation of neuroinflammation in the aSAH-injured brain by DPSC-CM. After 24 h of aSAH induction, the injured tissue was subjected to immunohistochemistry analysis for the measurement of Iba-1-positive cells. Induction of aSAH led to the substantial increase in Iba-1-positive cells over the injured brain that could be effectively ameliorated by DPSC-CM (panel **A**–**H**). To examine the involvement of IGF-1 in the DPSC-CM-mediated neuroprotective effect, IGF-1-neutralizing antibodies were co-administered with DPSC-CM. Neutralization of IGF-1 moderately blunted the DPSC-CM-mediated neuroprotective effect and drove the increase of Iba-1-positive cells (panel **I**). * *p* < 0.05, ** *p* < 0.01.

**Figure 6 ijms-20-03747-f006:**
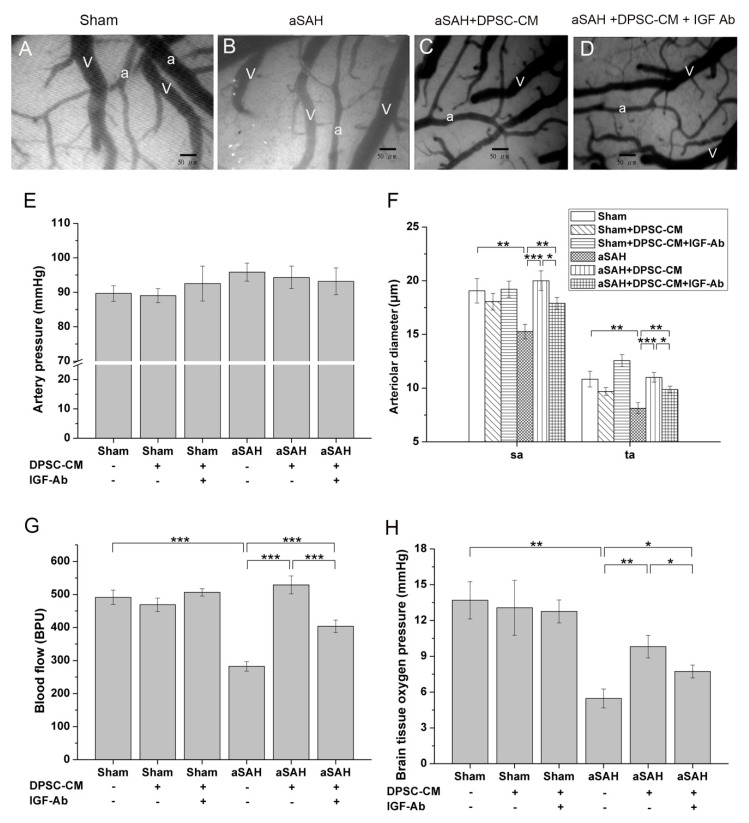
IGF-1 partially contributes to the beneficial effect of DPSC-CM on impaired brain microcirculation in rats with aSAH induction. After 24 h of induction of aSAH and treated with IGF-1 neutralizing antibody, as indicated, the image of microcirculation was captured as shown (**A**–**D**), indicating more constriction of secondary and terminal arterioles in the IGF-neutralized group than those of the DPSC-CM treatment group. Artery pressure (MAP) was not affected in all groups (**E**). Note that administration of IGF-1-neutralizing antibodies moderately blunted the DPSC-CM-mediated effects on vessel diameters (**F**), cortical blood flow (**G**), and the tissue oxygen pressure (H). * *p* < 0.05, ** *p* < 0.01, *** *p* < 0.001. “a”: artery. “v”: vein.

**Figure 7 ijms-20-03747-f007:**
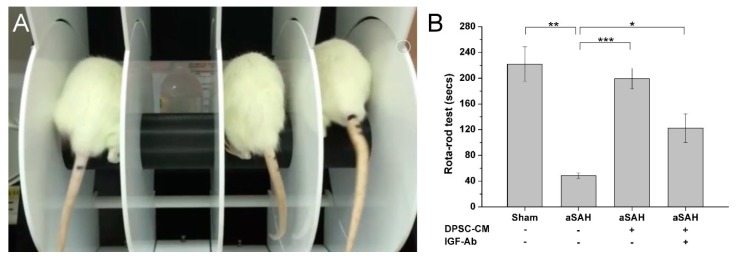
Improvement of neuro-cognitive impairments and motor dysfunctions in aSAH-injured rats by DPSC-CM. One week after the induction of aSAH, all experimental rats with indicated treatment were assigned for the Rotarod test (panel **A**). In aSAH-injured rats, induction of aSAH impaired the neurocognitive and motor functions, and treatment of DPSC-CM improved these impairments. Administration of IGF-1-neutralizing antibody moderately blunted the beneficial effect of DPSC-CM detected in the Rotarod test (panel **B**). * *p* < 0.05, ** *p* < 0.01, *** *p* < 0.001.

**Figure 8 ijms-20-03747-f008:**
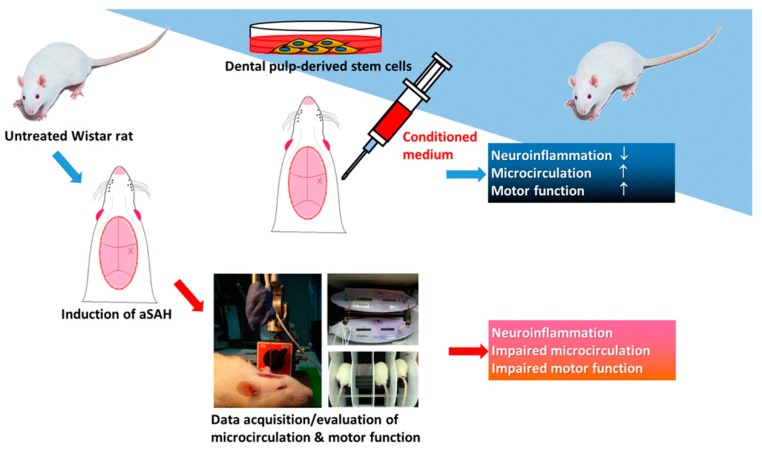
Illustrative scheme for the bioavailability of DPSC-CM that can improve neuroinflammation, impaired microcirculation, and motor dysfunction. Our established aSAH model has exhibited impaired microcirculation and motor dysfunction that recapitulates the clinical symptoms of aSAH. DPSC-CM carries promising efficacy that can improve impaired microcirculation and various neurological deficits in this aSAH model.

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
