# Peer review of "Dental Pulp Stem Cell-Derived Factors Alleviate Subarachnoid Hemorrhage-Induced Neuroinflammation and Ischemic Neurological Deficits"

_ijms, 2019, doi:10.3390/ijms20153747_

Round 1
Reviewer 1 Report
This is an interesting study that examines the benefit of DPSC derived factors in an experimental subarachnoid hemorrhage model in rats. Then study then goes on to identify IGF-1 as a key component of the beneficial response. The data clearly shows that IGF-1 is partly responsible for some of the benefit.
The main concern is the repeated use of the word neuroinflammation as the key mechanism. The increase and then decrease in iba-1 positive cells does not prove this is an neuroinflammatory pathway as other effects in the brain can cause iba-1 activation/increase in cell number by IHC. Without measuring changes in key inflammatory cytokines, it is presumptive to use the word neuroinflammatory. The use of the word ischemic is also problematic since it was not shown that true ischemia was affected by the DPSC/IGF-1. The rats showed an improved response, but the precise mechanism(s) are not definitively mediated by changes in ischemia alone.
The discussion should have less on cells and a bit more discussion about the CM from the cells.
The discussion should also have one or 2 more lines on IGF and vascular mechanisms, with appropriate references.
Minor concerns:
Incorrrect word "aneurysm" first sentence of abstract. Line 26
Need a concentration Line 152
The age of the rats should be noted
There are some scattered grammatical errors
Author Response
To Dear Reviewer1:
Comments and Suggestions for Authors
This is an interesting study that examines the benefit of DPSC derived factors in an experimental subarachnoid hemorrhage model in rats. Then study then goes on to identify IGF-1 as a key component of the beneficial response. The data clearly shows that IGF-1 is partly responsible for some of the benefit.
The main concern is the repeated use of the word neuroinflammation as the key mechanism. The increase and then decrease in iba-1 positive cells does not prove this is an neuroinflammatory pathway as other effects in the brain can cause iba-1 activation/increase in cell number by IHC. Without measuring changes in key inflammatory cytokines, it is presumptive to use the word neuroinflammatory. The use of the word ischemic is also problematic since it was not shown that true ischemia was affected by the DPSC/IGF-1. The rats showed an improved response, but the precise mechanism(s) are not definitively mediated by changes in ischemia alone.
l . We added this issue in discussion as following: About neuro-inflammation, microglia acts as the main immune defense scavenging cells in the brain, and it can against damaged neurons, debris or pathogens. While the activation of microglia occurred, the protective function can also give catastrophic results due to the secretion of nitric oxide , proteases, hydrogen peroxide and cytokines [27]. While microglia are the only type cells express Iba-1(ionized calcium binding adapter molecule 1) [28], Iba-1 expression is simultaneously up-regulated in activated microglia. Currently, enhanced Iba-1 expression had been observed in cerebral ischemia, traumatic brain injury and inflammation [29]. Here, we try to delineate the effects of neuroinflammation in SAH animal model with the quantification of Iba-1 stain.
2. We added this issue in discussion as following: While the delayed irreversible neurological deficits occurs several days after initial bleeding, it was believed that the clinical neurological deficits comes from the multiple and disseminated cerebral penumbra or ischemia (in severe form) [37]. In this study, the improvement in brain tissue oxygenation and blood flow had correlation to the motor improvement after DPSC-CM/IGF-1 treatment. Therefore, it could be presumed that the early intervention with DPSC-CM might give better clinical outcome due to decreased severity of cerebral ischemia or enhance cerebral perfusion.
The discussion should have less on cells and a bit more discussion about the CM from the cells.
3. We added this issue in discussion as following: The behaviors of MSCs are various, such as self-renew, differentiation, migration, even in aging and are accompanying with different compositions of paracrine for different purposes. The protocol of MSC-CM generation must be maintained in the same stage with similar potential as one of quality controls [14]. In this study, we made a purification of the constitutes in DPSC-CM with staged Tangential Flow Filtration in order to narrow down the spectrum of proteins in the DPSC-CM. Instead of using all the elements inside the DPSC-CM, we treat the SAH-rats with a relatively purified and quantified protein in order to delineate the anti-inflammatory effects.
The discussion should also have one or 2 more lines on IGF and vascular mechanisms, with appropriate references.
4. We added this issue in discussion as following: Previous study delineated how IGF-1 acts on vessels in injured neointima and tunica media in carotid stenosis, IGF-1 not only plays key role of proliferative or anti-apoptotic benefits, but also stabilized the endothelial progenitor cells in the carotid stenosis model [30].
Minor concerns:
Incorrrect word "aneurysm" first sentence of abstract. Line 26
The sentence had been corrected into: characterized by the extravasation of blood into the subarachnoid space caused by an intracranial aneurysm rupture
Need a concentration Line 152
That sentence had been corrected. The total protein concentration of DPSC-CM was 27 μg/ml.
The age of the rats should be noted
done
Reviewer 2 Report
Current manuscript by Te-Fu Chen et al., describes the beneficial role of dental pulp isolated mesenchymal stem cells (DPSC) in experimental rat model of Aneurysmal subarachnoid hemorrhage (aSAH). Authors observed that DPSC driven factors in the conditioned media were responsible for the beneficial functions, including reducing vasoconstriction, neuro-inflammation and improved cortical oxygenation in the injured brain. Moreover, Rotarod test revealed the aSAH-induced cognitive and motor impairments were significantly improved by the DPSC-CM administration. Additionally, antibody-mediated IGF-1 neutralization partially reversed the beneficial effect of MSC-derived conditioned media on microcirculation and tissue oxygenation, neuro-inflammation, and cognitive/motor functions, indicating that IGF-1 substantially contributed to the treatment effect induced by the conditioned media in aSAH-injured brain. This work is novel and has great potential to open new avenues in both mesenchymal stem cells based and cell free therapies in context of brain injuries. To my opinion, the experiments are well designed, performed and manuscript is well written with few typos which authors are encouraged to correct. Moreover, I have few concerns which authors should consider addressing before publishing this article.
- In Fig. 1, author did not use IgG controls in their FACS sorting experiments to validate their experiment. Also, in panel B author showed CD105A but did not describe in the text and in the legend. In panel A, where the microscopic images are shown, scale bars are missing.
- In fig. 2, panel B where author used pa, sa, ta it is not clear for readers what exactly they stand for. Thus, must be abbreviated or indicated clearly on fig. itself.
- Page 5, line 152, where the concentration of DPSC-CM is written it should be corrected to microgram as mentioned later in the manuscript.
- Fig. 5, where the images of microglia are shown, authors are suggested to provide more details which brain regions are shown in the images.
- Vogel et al., used in line 377 is missing in the reference list.
- Fig. 5 where IBA1 cells are quantified is somewhat very confusing as author showed two sham groups and second sham group is increased a lot in IBA1 positive count. This should be corrected.
- Statistical significance with various annotation like single star, double star etc. must be explained in figure legend/in method part.
- On page 12, authors have repeatedly used the term ‘Shame’ instead of ‘sham’ this must be corrected and also encouraged to properly check such errors throughout the manuscript.
- In their ELISA test, authors found second biggest constituent (39 %) of TIMP2, thus authors are encouraged at least to shed some light on this molecule in their discussion in context of aSAH or in general brain injuries.
Author Response
To Dear Reviewer2:
Current manuscript by Te-Fu Chen et al., describes the beneficial role of dental pulp isolated mesenchymal stem cells (DPSC) in experimental rat model of Aneurysmal subarachnoid hemorrhage (aSAH). Authors observed that DPSC driven factors in the conditioned media were responsible for the beneficial functions, including reducing vasoconstriction, neuro-inflammation and improved cortical oxygenation in the injured brain. Moreover, Rotarod test revealed the aSAH-induced cognitive and motor impairments were significantly improved by the DPSC-CM administration. Additionally, antibody-mediated IGF-1 neutralization partially reversed the beneficial effect of MSC-derived conditioned media on microcirculation and tissue oxygenation, neuro-inflammation, and cognitive/motor functions, indicating that IGF-1 substantially contributed to the treatment effect induced by the conditioned media in aSAH-injured brain. This work is novel and has great potential to open new avenues in both mesenchymal stem cells based and cell free therapies in context of brain injuries. To my opinion, the experiments are well designed, performed and manuscript is well written with few typos which authors are encouraged to correct. Moreover, I have few concerns which authors should consider addressing before publishing this article.
1. In Fig. 1, author did not use IgG controls in their FACS sorting experiments to validate their experiment. Also, in panel B author showed CD105A but did not describe in the text and in the legend. In panel A, where the microscopic images are shown, scale bars are missing.
The IgG control was used as the flow cytometry gating for CD marker. The red line of Figure 1B shows gating boundary line, distinguishing positive signal cells or not. Those IgG control were included in our experiments, but data not showed. The CD105 had been added in main text. And in panel A, the cell morphology picture had been replaced by clear photo with magnification of microscope.
2. In fig. 2, panel B where author used pa, sa, ta it is not clear for readers what exactly they stand for. Thus, must be abbreviated or indicated clearly on fig. itself.
The pa, sa, and ta mean primary arterioles (pa), secondary arterioles (sa) and terminal arterioles (ta), we had added the abbreviations into the legend and main text.
3. Page 5, line 152, where the concentration of DPSC-CM is written it should be corrected to microgram as mentioned later in the manuscript.
That sentence had been corrected. The total protein concentration of DPSC-CM was 27 μg/ml.
4. Fig. 5, where the images of microglia are shown, authors are suggested to provide more details which brain regions are shown in the images.
In the IHC staining, the tissue sections were acquired from brain cortex and detected the Iba-1 signal of aSAH-injured brain region.
5. Vogel et al., used in line 377 is missing in the reference list.
That reference, Vogel et al., had been corrected, cited in reference no.41.
6. Fig. 5 where IBA1 cells are quantified is somewhat very confusing as author showed two sham groups and second sham group is increased a lot in IBA1 positive count. This should be corrected.
The Figure 5 had been corrected. The second column is aSAH group without treating DPSC-CM and IGF-1 antibody.
7. Statistical significance with various annotation like single star, double star etc. must be explained in figure legend/in method part.
The marks of statistical analysis had been added into the legends.
8. On page 12, authors have repeatedly used the term ‘Shame’ instead of ‘sham’ this must be corrected and also encouraged to properly check such errors throughout the manuscript.
Those term “shame” had been checked and corrected to “sham” in whole article.
9. In their ELISA test, authors found second biggest constituent (39 %) of TIMP2, thus authors are encouraged at least to shed some light on this molecule in their discussion in context of aSAH or in general brain injuries.
A synergistic effect contributed to by all the cytokines from DPSC found in our data is possible; in particular, TIMP2 exists at a high level in DPSC-CM and is critical to the maintenance of tissue homeostasis. TIMP2 is an inhibitor for matrix metalloproteinase (MMP)-3 and 9, which have been shown to be highly related to higher risk of soft tissue trauma, such as Achilles tendinopathy. [38] As for functional study of TIMP2 in the neural system, it was remarkably addressed to be beneficial for noise-related cochlear injury [39]. Further studies will be required to elucidate the contribution of these bioactive constituents to the conditioned media-associated beneficial effect in the established aSAH model.
Thank you so much.
Reviewer 3 Report
The authors report on the effect of dental pulp-derived mesenchymal stem cells on microcirculation and neuroinflammation in a rat model of subarachnoid haemorrhage. Animal treated with SMC showed an improved microcirculation, a better tissue oxygenation and a decrease of neuroinflammatory reactions in the brain, which was found to be partially mediated by IGF-1.
Comments:
The authors should not use aneurysmal subarachnoid hemorrhage (aSAH) for their model as the bleeding is not caused by an aneurysm but an injection into the great cistern, which is usually not the place of origin of the bleeding. They should also note that they examine the early period of SAH which is should not be mixed up with the phase of vasospasm and DIND later in the course of the disease. They examine an early phase of disturbances in microcirculation which has already been described in patients undergoing early surgery and is a known phenomenon. These are different mechanisms in the disease and should not be mixed up.
Abstract
Lines 25,26: “Aneurysmal subarachnoid hemorrhage (aSAH), characterized by the extravasation of blood into intracranial aneurysm.” This statement is wrong. The blood extravasates from the aneurysm into the subarachnoid space.
Introduction
Line 54 -56: “Current treatments for aSAH include Tipple-H therapy (Hemodilution, Hypertension, and Hypervolemia) and surgical approaches such as craniotomy aneurysmal clipping or endovascular repairs”not use
Triple H-therapy instead of Tipple-H therapy
The sentence reveals that the authors are not really involved in treatment of SAH. Initial treatment for aSAH is surgery or endovascular occlusion of the aneurysm to prevent rebleeding. Triple-H therapy is used in the second phase of the disease when vasoconstriction may occur which usually starts from day 3-5 after bleeding. The sentence needs to be rephrased.
Results
The result section has in many paragraphs descriptions of material and methods and other statements not relevant for the description of the results (e.g. paragraph 2.2 lines 108 – 115 or paragraph 2.4 lines 164 – 175 (this is a report of a previous study)
In the legend of figure 2 DPSC-SM was applied by lumbar injection and by injection into the cisterna magna, in the text by injection into the cisterna magna (line 120-121).
Figure 3: What is meant by artery blood pressure? MAP, systolic or diastolic systemic blood pressure?
Abbreviations pa, sa and ta not explained. Asterisks are not explained. Statistical analysis not given in the legend.
The description of material and methods is insufficient. Please provide a clear experimental protocol what was done at what time. When were the injections performed? Was the injection of the stem
The measurement of the tissue oxygen is not described. Which method was used and where were the measurements taken? Were the animals mechanically ventilated during surgery and intravital microscopy? Were oxygen levels measured in the blood?
How were the diameters measured and how many vessels per region of interest were measured in each animal? Which diameter was measured in the constricting vessels: the constricted part or the non-constricted part? The term vessel diameter also includes venules, it might be better to speak of arteriolar diameters.
What was the actual number of animals used in total and in each group including the rotarod group and the immunostaining groups. What was the drop out rate? SAH is a severe disease and many animals die within the first 24h. There is also injury to the brain or blood ouzing onto the brain surface during cranial window preparation which leads to exclusion of animals.
At what day was the Rotarod test performed?
The legend of figure 6 is not correct.
Figures 6a and b have already been published previously (Wang et al. Sci Rep 2018, 8, (1), 13315. Figures 2a and b).
Author Response
To Dear Reviewer3:
Comments and Suggestions for Authors:
The authors report on the effect of dental pulp-derived mesenchymal stem cells on microcirculation and neuroinflammation in a rat model of subarachnoid haemorrhage. Animal treated with SMC showed an improved microcirculation, a better tissue oxygenation and a decrease of neuroinflammatory reactions in the brain, which was found to be partially mediated by IGF-1.
Comments:
The authors should not use aneurysmal subarachnoid hemorrhage (aSAH) for their model as the bleeding is not caused by an aneurysm but an injection into the great cistern, which is usually not the place of origin of the bleeding. They should also note that they examine the early period of SAH which is should not be mixed up with the phase of vasospasm and DIND later in the course of the disease. They examine an early phase of disturbances in microcirculation which has already been described in patients undergoing early surgery and is a known phenomenon. These are different mechanisms in the disease and should not be mixed up.
Abstract
Lines 25,26: “Aneurysmal subarachnoid hemorrhage (aSAH), characterized by the extravasation of blood into intracranial aneurysm.” This statement is wrong. The blood extravasates from the aneurysm into the subarachnoid space.
The sentence had been corrected : characterized by the extravasation of blood into the subarachnoid space caused by an intracranial aneurysm rupture
Introduction
Line 54 -56: “Current treatments for aSAH include Tipple-H therapy (Hemodilution, Hypertension, and Hypervolemia) and surgical approaches such as craniotomy aneurysmal clipping or endovascular repairs”not use
Triple H-therapy instead of Tipple-H therapy
This typing error had been corrected with Triple H-therapy.
The sentence reveals that the authors are not really involved in treatment of SAH. Initial treatment for aSAH is surgery or endovascular occlusion of the aneurysm to prevent rebleeding. Triple-H therapy is used in the second phase of the disease when vasoconstriction may occur which usually starts from day 3-5 after bleeding. The sentence needs to be rephrased.
This sentence had been rephrased as:” Current treatments for aSAH related vasoconstriction include Triple-H therapy (Hemodilution, Hypertension, and Hypervolemia) in order to keep adequate cerebral perfusion and initial surgical approaches such as craniotomy aneurysmal clipping or endovascular repairs aim to prevent rebleeding.
Results
The result section has in many paragraphs descriptions of material and methods and other statements not relevant for the description of the results (e.g. paragraph 2.2 lines 108 – 115 or paragraph 2.4 lines 164 – 175 (this is a report of a previous study)
Those descriptions were intended to illustrate the results of subsequent experiments in this study. We had reduced the narrative and demonstrated in material and methods or discussion
In the legend of figure 2 DPSC-SM was applied by lumbar injection and by injection into the cisterna magna, in the text by injection into the cisterna magna (line 120-121).
The description in the legend was wrong. The correct administration is injected via foramen magnum. The main text had been corrected.
Figure 3: What is meant by artery blood pressure? MAP, systolic or diastolic systemic blood pressure?
In our study, the artery blood pressure indicates the mean arterial pressure (MAP), calculating by one-third systolic blood pressures (SP) plus two-thirds diastolic blood pressures (DP).
Abbreviations pa, sa and ta not explained. Asterisks are not explained. Statistical analysis not given in the legend.
The pa, sa, ta were described in the legend, that mean primary arterioles (pa), secondary arterioles (sa) and terminal arterioles (ta), we had added the abbreviations into the legend and main text. And the marks of statistical analysis had been added into the legends.
The description of material and methods is insufficient. Please provide a clear experimental protocol what was done at what time. When were the injections performed? Was the injection of the stem
The method had been added to paragraph 4.5 of main text. The conditioned medium of DPSC and neutralizing antibodies of IGF-1 were injected into cisterna magnum 10 minutes before aSAH.
The measurement of the tissue oxygen is not described. Which method was used and where were the measurements taken? Were the animals mechanically ventilated during surgery and intravital microscopy? Were oxygen levels measured in the blood?
The method had been added to paragraph 4.2 of main text. The extent of tissue perfusion and PbtO2 in the brain cortex were measured using an OxyLite 2000E detector and an OxyFLO 2000E detector (Oxford Optronic Ltd, England) (Childs Nerv Syst 2010, 26, 419–430). The probe was fixed on a stereotactic frame (Stoelting, Wood Dale, IL, USA), and the tip was inserted to a depth of 2 mm underneath the brain surface to start recording.
How were the diameters measured and how many vessels per region of interest were measured in each animal? Which diameter was measured in the constricting vessels: the constricted part or the non-constricted part? The term vessel diameter also includes venules, it might be better to speak of arteriolar diameters.
In our study, after the craniotomy and dura opening, we used a dissection microscope equipped with a capillary videoscope to monitor the cortical microcirculations. With this setup, the cortical microcirculations including the main arterioles and venules could be clearly observed. Around 1 to 2 main arterioles could be found in the craniotomy site, and each arteriole was branched into primary arterioles (pa), secondary arterioles (sa), and terminal arterioles (ta). Therefore, we actually measured the arteriole diameter of primary (pa), secondary (sa) and terminal (ta) arterioles in the brain microcirculation. That had been described in paragraph 2.2 of main text. Meanwhile, we had corrected the marks in figure 3 and 6.
What was the actual number of animals used in total and in each group including the rotarod group and the immunostaining groups. What was the drop out rate? SAH is a severe disease and many animals die within the first 24h. There is also injury to the brain or blood ouzing onto the brain surface during cranial window preparation which leads to exclusion of animals.
In the aSAH rat model we established, that recapitulates neural damage and impaired microcirculation [20]. The mortality rate was very low, only 3%. Because of we want to observe the vasoconstriction of secondary arterioles (sa), and terminal arterioles (ta) in the brain, not the main arteriole, that would achieve the effect of recapitulates neural damage and impaired microcirculation. Therefore, we just injected fewer autologous blood one time into the cisterna magna. This is the reason for achieving lower mortality.
At what day was the Rotarod test performed?
The Rotarod test was performed after one week of aSAH induction. We had added the description in paragraph 4.6 of main text.
The legend of figure 6 is not correct.
Figures 6a and b have already been published previously (Wang et al. Sci Rep 2018, 8, (1), 13315. Figures 2a and b).
Figure 6A and B were mistakes. We uploaded the wrong version of our data. Figure 6A and B had been replaced by the correct figure in the new version of our manuscript.
Thank you so much.
Round 2
Reviewer 1 Report
The revised wording is confusing. Although the authors have attempted to respond to many of the reviewers concerns, the grammar does not lend itself to understanding the science.
Reviewer 2 Report
Authors have now responded to most of the queries raised by reviewer thus current manuscript can now be accepted in its form.
Reviewer 3 Report
No further comments.